# Two-Pore Channels Regulate Inter-Organellar Ca^2+^ Homeostasis in Immune Cells

**DOI:** 10.3390/cells11091465

**Published:** 2022-04-26

**Authors:** Philip Steiner, Elisabeth Arlt, Ingrid Boekhoff, Thomas Gudermann, Susanna Zierler

**Affiliations:** 1Institute of Pharmacology, Medical Faculty, Johannes Kepler University Linz, 4020 Linz, Austria; philip.steiner@jku.at; 2Walther Straub Institute of Pharmacology and Toxicology, Medical Faculty, Ludwig Maximilians University Munich, 80336 Munich, Germany; elli.arlt@googlemail.com (E.A.); ingrid.boekhoff@lrz.uni-muenchen.de (I.B.); thomas.gudermann@lrz.uni-muenchen.de (T.G.)

**Keywords:** TPC, two-pore channel, TPC1, immune cell, mast cell, calcium, Ca^2+^, endosome, lysosome, endolysosome, anaphylaxis, histamine

## Abstract

Two-pore channels (TPCs) are ligand-gated cation-selective ion channels that are preserved in plant and animal cells. In the latter, TPCs are located in membranes of acidic organelles, such as endosomes, lysosomes, and endolysosomes. Here, we focus on the function of these unique ion channels in mast cells, which are leukocytes that mature from myeloid hematopoietic stem cells. The cytoplasm of these innate immune cells contains a large number of granules that comprise messenger substances, such as histamine and heparin. Mast cells, along with basophil granulocytes, play an essential role in anaphylaxis and allergic reactions by releasing inflammatory mediators. Signaling in mast cells is mainly regulated via the release of Ca^2+^ from the endoplasmic reticulum as well as from acidic compartments, such as endolysosomes. For the crosstalk of these organelles TPCs seem essential. Allergic reactions and anaphylaxis were previously shown to be associated with the endolysosomal two-pore channel TPC1. The release of histamine, controlled by intracellular Ca^2+^ signals, was increased upon genetic or pharmacologic TPC1 inhibition. Conversely, stimulation of TPC channel activity by one of its endogenous ligands, namely nicotinic adenine dinucleotide phosphate (NAADP) or phosphatidylinositol 3,5-bisphosphate (PI(3,5)P_2_), were found to trigger the release of Ca^2+^ from the endolysosomes; thereby improving the effect of TPC1 on regulated mast cell degranulation. In this review we discuss the importance of TPC1 for regulating Ca^2+^ homeostasis in mast cells and the overall potential of TPC1 as a pharmacological target in anti-inflammatory therapy.

## 1. Introducing Two-Pore Channels

Two-pore channels (TPCs) are intracellular voltage- and ligand-gated cation channels in eukaryotic cells [1]. TPCs play a special role in numerous immunological and (patho-)physiological processes [2,3,4] TPCs have two functional members in the human genome (TPC1 and TPC2) and belong to a novel class of non-selective cation channels [5]. TPCs have two subunits and each subunit consists of 12 transmembrane helices with two pore domains between the 5th and 6th segment or the 11th and 12th segment, respectively (Figure 1a). The formation of dimers creates an asymmetrical pore, which is formed by neutral amino acids; thus allowing Na^+^, Ca^2+^, and H^+^ to penetrate the pore [6,7,8]. Figure 1a also shows the glycosylation sites on the luminal side (in red) that were previously discovered [9].

TPCs are subdivided into TPC1, TPC2, and TPC3. Compared to TPC1 and TPC2, TPC3 was not found in human, primate, or rodent cells [10]. However, TPC3 was found in the plasma membrane of cortical granules of starfish oocytes [11]. The protein structure of TPC1 was the first to be resolved in *Arabidopsis thaliana* by means of X-ray crystallography [1,12]. Only recently, the 3D structure of TPC1 and TPC2 was shown for *Mus musculus* and *Homo sapiens*, respectively, using cryo-electron microscopy. A representative cryo-electron microscopy (EM) derived three-dimensional structure of the source organism *Mus musculus* in the apo state (inactive) is shown in Figure 1b, see also [13,14].

In this review we focus on the cation channel TPC1 and its function in mast cells, regulating intracellular Ca^2+^ signals. We further discuss the potential role of TPC1 as a pharmacological target for a wide range of diseases, from anaphylaxis [2] to viral infections [15,16].

## 2. Mast Cells and Basophil Granulocytes Regulate Anaphylaxis

Mast cells and basophils are innate leukocytes that both develop from hematopoietic stem cells. Mast cells can be found in almost all organs and are often located near neurons, blood vessels, and lymphatic vessels to transmit local inflammatory signals [17]. The intracellular structure of mast cells is unique as it contains a large number of granules, which are filled with multi-potent hormones, such as histamine, heparin, leukotrienes, many cyto- and chemokines, and growth factors [18,19,20]. Mast cells represent an important part of the innate immune system, which is essential for the defense against pathogens, such as viruses and bacteria, but also for wound healing. They are key effector cells in allergies and anaphylactic responses [2]. Activation takes place via the binding of immunoglobulin E (IgE) to the Fcε-receptor (FcεR) and subsequent high affinity binding to the respective allergen, resulting in crosslinking of two or more FcεR molecules [20,21]. During this process, inflammatory mediators, such as histamine, are released through degranulation. Numerous factors that favor an excessive release of histamine and heparin from the mast cell are the triggers of allergies, such as allergic asthma [22,23]. However, mast cells serve far more physiological functions than the well-known release of histamine [24,25,26]. In addition to mast cells, basophil granulocytes are also of great importance during anaphylaxis and allergic reactions, and consequently for the therapy of, for example, asthma [27]. In fact, there are some properties of basophils that are also found in mast cells. For example, like mast cells, basophils also contain characteristic granules in the cytoplasm [28]. Furthermore, basophils can also release mediators, such as histamine and cysteinyl leukotrienes [29,30]. Unlike mast cells, basophils primarily reside in the blood [30].

## 3. Inter-Organellar Ca^2+^ Homeostasis in Mast Cells

Allergen exposure and binding to IgE pre-coupled to FcεR in the plasma membrane of a mast cell activates phospholipase C (PLC); thus leading to the release of inositol 1,4,5-triphosphate (IP_3_; depicted in Figure 2) and diacylglycerol (DAG). IP_3_ then binds to the IP_3_ receptor (IP_3_R), which is located in the endoplasmic reticulum (ER) membrane. The ER is one of the most important organelles for the regulation of intracellular Ca^2+^ and the largest Ca^2+^ store in the cell [31]. The binding of IP_3_ to IP_3_R results in a release of Ca^2+^ from the ER, which, in turn, triggers a regulated exocytosis of secretory vesicles and the accompanying release of histamine from the cell. It was recently demonstrated that this process is fine-tuned by TPC1 in the endolysosomal membrane [2]. Therefore, the endolysosome has to be in close proximity to the ER (Figure 2, left panel; contact sites). The Ca^2+^-permeable TPC1 can regulate the level of ER-Ca^2+^ by facilitating the release and uptake (likely indirectly) of Ca^2+^ from and into the endolysosome, respectively, and thus support the regulated exocytosis of histamine containing vesicles (Figure 2, left panel). The endolysosomal compartment itself has a homeostatic function, tightly regulating the ER Ca^2+^ concentration, and thereby the reactivity of mast cells. This suggested homeostatic crosstalk is dependent on TPC1 function in that TPC1 activity triggers a feed-forward loop causing the opening of IP_3_-receptors. The local increase of Ca^2+^ is then triggering its uptake into endolysosomes; however, the exact molecular mechanisms underlying this uptake process remain elusive and warrant further investigations. With the lack of crosstalk of these two storage compartments, the ER is “overly” filled with Ca^2+^. However, in this context it is not yet fully understood how Ca^2+^ is actively taken up into the endolysosome and whether additional transporters are involved. If TPC1 is blocked genetically or through pharmacological inhibition by means of, for example, the plant alkaloid tetrandrine or by antagonizing its activation via nicotinic acid adenine dinucleotide phosphate (NAADP) using *trans*-Ned-19 [32], there is no regulation of intracellular Ca^2+^ by means of TPC1 in the endolysosome [2]. When the signal cascade is then activated by IP_3_ due to repeated allergen exposure, there is an increased release of the previously augmented ER-Ca^2+^ due to the aforementioned inhibition of TPC1. This, in turn, results in enhanced exocytosis, as shown in Figure 2 (right panel) and, consequently, in an anaphylactic reaction.

## 4. TPC Regulation by NAADP and PI(3,5)P_2_

It is well known that TPCs can be activated by NAADP (see Figure 2), a Ca^2+^ mobilizing agent [33,34,35]. For cytotoxic T cells it has been shown that NAADP application triggers the release of Ca^2+^ from the (endo-)lysosome, which subsequently provokes a secretion of perforin and granzyme from the secretory granules into the extracellular space [36,37]. In previous studies, binding sites of NAADP were examined [38,39]. There, it was found that NAADP most likely cannot bind directly to TPCs, but indirectly via accessory proteins. Such an accessory protein was recently identified [40,41]. The Jupiter microtubule associated homolog 2 (JPT2), also known as Sm-like protein LSM12, was suggested to be such an interaction partner for NAADP and TPCs.

Previously, it was also described that TPCs can be activated via PI(3,5)P_2_ (phosphatidylinositol 3,5-bisphosphate; see Figure 2) and in this context were considered as voltage-dependent Na^+^ channels [8,13,42,43]. This has been assumed because activation of TPCs via PI(3,5)P_2_ rather leads to a Na^+^ than a Ca^2+^ flux [7,13,42,44].

In this context it is vital to mention a recently published work by Gerndt et al. [7], who found that ion selectivity of the TPC2 in primary murine macrophages depends on the activating ligand, thereby differentially promoting lysosomal function. Therein, the authors show that activation of TPC2 by means of PI(3,5)P_2_ leads to an increase in lysosomal exocytosis, whereas activation by means of NAADP decreases lysosomal exocytosis due to a change in vesicular pH. This has been previously described for NAADP and can probably be attributed to the agonist-specific effect on proton permeability [45,46]. However, on the same subject, but in this case using mast cells as an exocytotic model system, it was found by Arlt et al. [2] that activation of TPC1 by either NAADP or PI(3,5)P_2_ alone (see Figure 2) does not lead to an enhanced degranulation of mast cells. Neither TPC1-deficient nor wild-type mast cells showed a change in their membrane surface area upon perfusion. However, upon simultaneous activation of G-proteins by the nonhydrolyzable analogue of GTP, GTPγS, degranulation was even reduced, suggesting that the timely release of local Ca^2+^ from the endo-lysosomes is key for regulated exocytosis. Consequently, when TPC1 was inhibited with the plant alkaloid tetrandrine or with the NAADP antagonist *trans* Ned-19, exocytosis was enhanced due to an increase in cytosolic Ca^2+^ released from the ER. The fact that the debate about endogenous ligands of TPCs, namely NAADP and PI(3,5)P_2_, is relatively controversial was previously taken up in several articles [42,47,48,49,50]. In Ruas et al. [48] it was suggested that TPCs are important for the conduction of both cations (Ca^2+^ and Na^+^) and that activation is possible via both NAADP and PI(3,5)P_2_. It was assumed that this could not always be clearly demonstrated in the past due to inadequate mouse models in which the TPC function was sometimes unintentionally preserved [47,48].

When distinguishing between plant and mammalian TPC channels, the latter have been suggested as potential receptors for NAADP [33,34,51]. However, there is also increasing evidence in recent studies that TPCs are Na^+^ selective channels and are also activated by PI(3,5)P_2_ instead of NAADP [42,50]. A decisive step forward in this matter came from the work of Guo et al. [8], who performed an electrophysiological characterization and comparison of the ion selectivity properties of plant AtTPC1 (*Arabidopsis thaliana*) and mammalian HsTPC2 (*Homo sapiens*). By the implementation of structure-directed mutagenesis, the nonselective AtTPC1 was transformed into a Na^+^ selective channel similar to HsTPC2. This identified the most important filter residues that are central to the definition of ion selectivity of TPCs. Ultimately, however, there are still some gaps in this context that must be filled with future experiments, especially since PI(3,5)P_2_ can regulate TPC1 in addition to NAADP and this more likely leads to a Na^+^-specific current. The recent discovery of two promising agonists for TPC2, which seem to selectively trigger Na^+^- or Ca^2+^-conductance, might help to shed light onto this conundrum [7]. The first agonist, named TPC2-A1-P, seems to be responsible for selective Na^+^ currents. The second agonist, termed TPC2-A1-N, in turn triggers an increased Ca^2+^ signal. Based on these results, it has already been speculated that TPC2-A1-N could be an imitator of NAADP, whereas TPC2-A1-P probably mimics PI(3,5)P_2_ [7]. However, the question then arises as to why, in the case of TPC-A1-N, no accessory proteins, such as JPT2/LSM12, are required, as it has been suggested for NAADP [40,41]. Therefore, future ultrastructural experiments are necessary, which may provide further insights into the exact molecular mechanism of channel activation.

## 5. TPCs as Pharmacological Targets

The plant alkaloid tetrandrine was mentioned several times before as an inhibitor for TPC1 and TPC2 [2,52,53]. In a recently published hypothesis paper on SARS-CoV-2, the pharmacological assumption was made that the inhibition of TPC2 by means of tetrandrine represents a potential approach for the therapy of a corresponding virus infection [54]. However, the authors also pointed out that a potential utilization of tetrandrine in SARS-CoV-2 infection naturally needs to be guided by available experimental evidence as well as clinical experience.

In another study [55], TPC2 again moved into the scientific focus with regard to the virus uptake of SARS-CoV-2. Using TPC2 inhibitors, it was possible to detect a reduced virus entry and, based on this fact, TPC2 was suggested as a potential target for antiviral therapy against SARS-CoV-2 [16,55]. Both hypotheses [54,55] are in line with an earlier study in which the treatment of a common cold with tetrandrine suppressed the replication of human coronaviruses (HCoV) [56]. In addition, the flavanone naringenin was also recently investigated as a potential drug for the treatment of SARS-CoV-2 [57]. Similar to tetrandrine [54], naringenin should also serve as an inhibitor in the course of endolysosomal virus uptake, and thus prevent the release of the virus genome from the endolysosome via TPCs [57,58]. In addition to tetrandrine, further compounds have only recently been identified that have an even more specific effect on TPCs, especially TPC2, and are less toxic than tetrandrine [3].

The selective antagonist of the Ca^2+^-activating messenger NAADP, *trans* Ned-19 (see Figure 2), has also been previously described as a promising modulator for TPCs [32]. It was recently reported that TPCs are associated with the release of Ca^2+^ from vesicles into the cytoplasm and that the vesicular pH and the vesicle circulation are also affected [5]. This is also directly related to virus trafficking. For example, it has been shown that TPCs play an important role in the trafficking of viruses, such as Ebola (EBOV) and Mers (MERS-CoV) or bacterial toxins such as cholera toxin. By inhibiting TPCs, a reduction in the infectivity of these viruses and bacteria has already been demonstrated [15,52,59].

In addition to the abovementioned studies on TPC inhibitors, there are also TPC agonists which have great potential for pharmacological research, namely the two abovementioned novel small molecule agonists for TPC2, which were found during a high-throughput screening [7]. Besides the ability to trigger ion selectivity of these new TPC2 agonists, TPC2-A1-P and TPC2-A1-N have distinct binding sites suggesting alternate mode of actions. Furthermore, the authors found that the binding sites of the two agonists are not detectable in TPC1 nor in the mucolipin subfamily of the transient receptor potential cation channel (TRPML). TPC2-A1-N and TPC2-A1-P could, therefore, be site-selective and specific TPC2 agonists that may have enormous potential for future pharmacological applications. An overview of the discussed pharmacological modulators of TPCs is shown in Table 1.

## 6. Outlook

In the past few years, TPCs have moved more and more into the focus of interest, especially for immunological and pharmacological questions [2,3,54]. On the basis of the pharmacological modulation of TPC1 in mast cells, it was possible to gain important insight into a potential therapy to treat allergic hypersensitivity [2]. The latter study was only recently discussed in a comment article [61]. Herein, the remaining question of how the regulation of intracellular Ca^2+^ via TPC1 actually takes place was highlighted. The comment [61], therefore, also referred to a study [34] in which a knockdown of TPC1 in the human breast cancer cell line SKBr3 revealed no difference in the IP_3_ induced Ca^2+^ release from the ER. Accordingly, the authors [61] concluded that the outcome of targeting TPC1, pharmacologically or genetically, could be cell-type specific. However, one must also consider that SKBr3 cells are tumor cells and, for that reason alone, the cells might react differently in ER-Ca^2+^ release after IP_3_ induction. Nevertheless, there are still some gaps in this context that need to be filled with additional experiments.

TPCs also play an important role in the current SARS-CoV-2 pandemic [16,55,57]. There are promising inhibitors that seem to suppress the release of the virus genome during the endolysosomal pathway [54,55,57]. Nonetheless, further investigations are also required here to validate the pharmacological effectiveness of these drugs for virus infections.

Apart from the well-documented physiological and molecular biological research, there are important gaps that could be closed with high-resolution, ultrastructural investigations. An example for this would be the interactions of organelles at the biomembrane level, the exact intracellular localization of TPCs, or the identification and subcellular visualization of possible NAADP binding partners. As structure follows function and vice versa, electron microscopic methods could answer ultrastructural questions on the basis of the already broad physiological and molecular biological knowledge of TPCs. Contact sites between ER and endolysosomes, as indicated in Figure 2, were previously assessed with 2D EM methods and analyzed with regard to the Ca^2+^ movement and distribution between the ER and the endo-lysosomal system [62]. Here, one could evaluate 3D tomograms in future ultrastructural experiments in order to get an even better understanding of the interaction of the three-dimensional organelles. If one wants to exhaust the EM method spectrum even further, analytical methods, such as electron energy loss spectroscopy (EELS), electron spectroscopic imaging (ESI), and energy dispersive X-ray analysis (EDX), would have to be mentioned with regard to Ca^2+^ homeostasis and signaling in order to get a better insight into the elementary distribution. Additionally, immunocytochemical determination of TPCs, as it has already been carried out before [63], or the identification of ER and endolysosomal contact sites (see also Figure 2) could sketch a more accurate picture around TPCs [63].

While 2D and 3D electron microscopic methods including element analysis and immunocytochemistry could clarify how organelles interact to maintain Ca^2+^ homeostasis, a combination with electrophysiological, cell and molecular biological methods could help in answering open immunological and (patho)physiological questions. Consequently, the potential of TPCs as pharmacological targets for the treatment of diseases, such as anaphylaxis or infectious diseases, has to be investigated further and more comprehensively. Furthermore, one should consider that expression and function of TPCs could be cell type-specific, as this has only recently been discussed [61]. In this context, very little is known about TPCs in adaptive immune cells, such as T cells [4]. For example, the role of TPCs in primary CD4^+^ (T-helper) cells and CD8^+^ T- Lymphocytes (CTLs) needs to be investigated in more detail. The precise localization of TPCs in various intracellular vesicles, such as the granules in granulocytes and mast cells as well as in lytic vesicles of Natural Killer (NK) cells or CTLs, is still unclear. In this context, there are many open intriguing questions that need to be answered.

## Figures and Tables

**Figure 1 cells-11-01465-f001:**
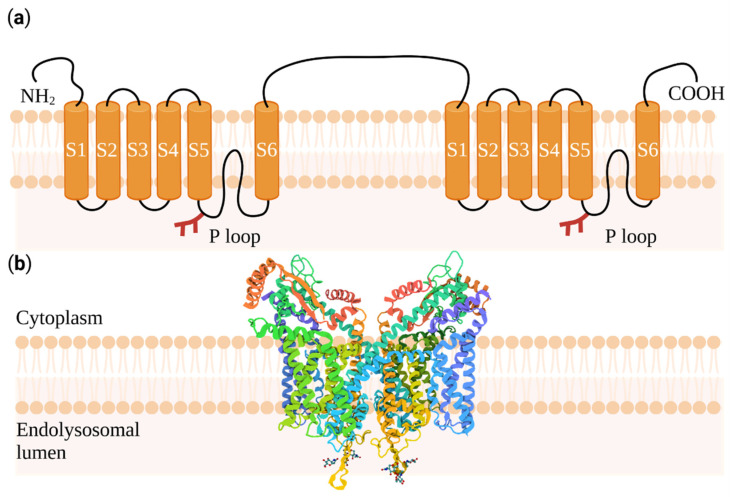
Structure of TPC1 in biological membranes. (**a**) The scheme illustrates TPC1 as a homodimer where each subunit is composed of six transmembrane domain helices. The luminal red appendages show the glycosylation sites on each P-loop. Each subunit forms a pore region between the 5th and 6th segment (S); (**b**) the cryo-EM structure shows TPC1 of *Mus musculus* at a resolution of 3.5 Å in the apo state. For the illustration, the protein data bank (PDB) code 6C96 was applied. Created with BioRender.com, accessed on 29 March 2022.

**Figure 2 cells-11-01465-f002:**
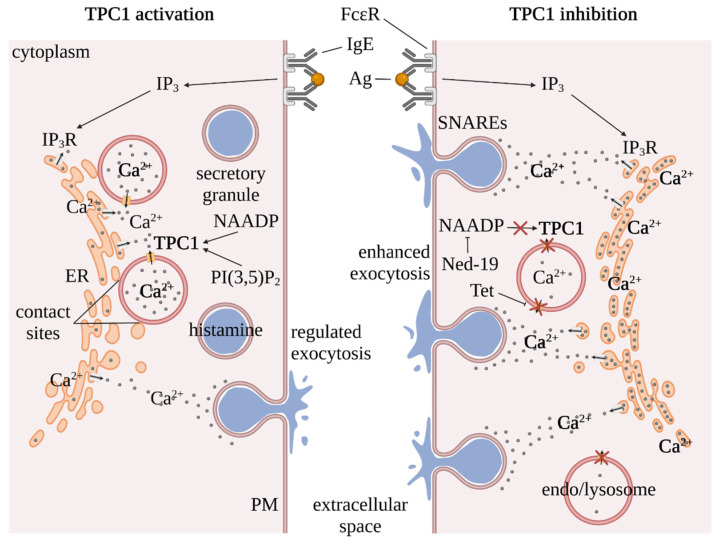
Illustration of the regulation of Ca^2+^ homeostasis via TPC1 in mast cells. The schematic drawing on the left signifies the role of TPC1 in organellar Ca^2+^ homeostasis upon IgE-induced exocytosis. FcεR is activated by IgE- and triggers phospholipase C, which, in turn, creates IP_3_ and DAG (DAG is not depicted). Then, IP_3_ binds to IP_3_R, which is located in the ER membrane and releases ER-Ca^2+^. Interestingly, a reciprocal upregulation of Ca^2+^ outflow from the ER and the endo-lysosomal system has been identified, which can be regulated by a TPC1-triggered uptake of Ca^2+^ into the endolysosome; thereby leading to controlled histamine release. Important agonists for TPC1 for this indirectly controlled outflow of Ca^2+^ from the ER are PI(3,5)P_2_ and NAADP (**left panel**). If TPC1 is inhibited pharmacologically with, for example, tetrandrine or indirectly with *trans* Ned-19 via NAADP-antagonization, TPC1-controlled transport of Ca^2+^ into the endo-lysosomal system becomes deregulated, which subsequently leads to an accumulation of Ca^2+^ within the ER (**right panel**). Consequently, FcεR stimulation and the followed activation of the signaling cascade triggers an elevated ER-Ca^2+^ outflow which, hence, then results in an enhanced exocytosis of histamine. PM: plasma membrane; IgE: Immunoglobulin E; FcεR: high affinity receptor of the Fc region of IgE; ER: endoplasmic reticulum; IP_3_R: inositol 1,4,5-triphosphate receptor; NAADP: nicotinic acid adenine dinucleotide phosphate; PI(3,5)P_2_: phosphatidylinositol 3,5-bisphosphate; SNARE: soluble NSF attachment protein receptor; red crosses: inhibition of TPC1 by Tet or Ned-19 (via NAADP). Created with *BioRender.com*, accessed on 29 March 2022.

**Table 1 cells-11-01465-t001:** Overview of the discussed pharmacological modulators in this review that act directly or indirectly on TPCs.

Name	Mode of Modulation	References
Naringenin	Inhibitor	[60]
Tetrandrine	Inhibitor	[15]
Ned-19	Inhibitor	[32]
TPC2-A1-N	Activator	[7]
TPC2-A1-P	Activator	[7]
SG-005	Inhibitor	[3]
SG-094	Inhibitor	[3]

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
