# Peer review of "Two-Pore Channels Regulate Inter-Organellar Ca2+ Homeostasis in Immune Cells"

_cells, 2022, doi:10.3390/cells11091465_

Round 1

Reviewer 1 Report

This manuscript reviews the current knowledge on the functional role of TPC channels in endolysosomes with special emphasis on innate immune cells and provides insight into the pharmacological and pathophysiological implications of this information also regarding to the potential use of TPC channels as targets to fight SARS-CoV-2 infection. The manuscript is concise and reads well, and offers a clear and equilibrated image of the controversial aspects of the topic. I have only some minor language and presentation suggestions that could help the non-specialist readers.

Line 24 were instead of we?

Line 39. Figure 1a also shows the glycosylation sites on the sour luminal side (in red) [8].

Line 43. I am not sure if “ultrastructure” is the most accurate word when referring to protein structure.

Figure 1. In the PDF the monochrome representation of the protein structure does not allow to appreciate the different features.

Lines 74-75. The sentence starting “One essential protein that seems” appears out of context here.

Line 133 “was proposed or suggested instead of “was expected”?

Lines 161-163. The statement attributed to ref. 44 is actually a quotation from reference 45 in that paper.

Reference 52 is a correction to a 2020 paper (https://doi.org/10.1038/s41467-020-15562-9).

Reviewer 2 Report

General Comments:

This review from Steiner et al. is dealing with the role of two-pore channels (TPC) in the regulation of Ca2+ signalling in immune cells. This manuscript is well written and provides important highlights in the physiology of mast cells. I have only minor comments in order to improve this elegant work.

Minor comments:

  1. Abstract, lines 22-26: This sentence seems confuse, please check.
  2. Figure 2 and related text: It is not clear how is the Ca2+ flux through TPC channels. In the figure 2 is shown a release of Ca2+ out of the endolysosome that is mediated by TPC channels whereas it is indicated that TPC channels are involved in the absorption and release of Ca2+ into the endolysosome (line 95). Please clarify. Moreover, the arrows in the endolysosome are very small and should be larger.
  3. The fifth paragraph could be completed with a table containing information on the known pharmacological modulators of TPC.

Reviewer 3 Report

In this review, the authors, who themselves have previously contributed to the elucidation of TPC function including that in mast cells, first give a general overview of TPC channels and the function of innate leukocytes, followed by interorganellar Ca2+ signaling and then in more detail, the regulation of TPC function and its pharmacological modification. This is a timely review on a very important topic. The authors did well in concisely summarizing a complex issue, however at some points at the cost of omitting relevant points for discussion or reduced clarity.

In the following some comments and suggestions that might improve the review:

  • The first paragraph of the introduction is not very clear. “TPCs have two functional members in the genome”: which ones, what are the gene names? In what sense do they belong to “a novel class of ion channels”? “TPCs have two subunits”, meaning that the TPC channel is formed by TPC dimers? What does “the sour luminal side” mean?
  • Figure 1b shows TPC1 and depicts the lysosomal lumen. Is TPC1 lysosomal?
  • Lines 94-96: This deserves more explanation: activation of TPC1 leads to Ca2+ efflux under basal conditions, but Ca2+ influx upon release of ER Ca2+? How does influx of Ca2+ into TPC1-positive vesicles affect ER Ca2+, considering the much larger cytosolic space?
  • Lines 101-102: what do the authors mean by “co-regulation”
  • Lines 103-104: it is not clear from the text why ER-Ca2+ would now have been previously augmented. Is the SERCA more active or ER-Ca2+ release channels altered?
  • Line 133: why “expected”?
  • Lines 137ff: this is an important point about TPCs: their permeabilities for Na+ and Ca2+ and the related debate in the literature. This issue should be explained in more detail in this review.
  • Line 141: explain why NAADP may affect pH.
  • Lines 140-141 and lines 143-147 repeat the same contents.
  • Line 164: meant “former” instead of “latter” or is the following (again) on mammalian TPCs?
  • Line 165: not “receptors” as another protein may be the receptor (as the authors also state).

Round 2

Reviewer 3 Report

The authors have sufficiently addressed all my concerns.